# Effects of Functional Partial Body Weight Support Treadmill Training on Mobility in Children with Ataxia: A Randomized Controlled Trial

**DOI:** 10.3390/jfmk10020123

**Published:** 2025-04-06

**Authors:** Alexandra Lepoura, Sofia Lampropoulou, Antonis Galanos, Marianna Papadopoulou, Georgios Gkrimas, Magda Tziomaki, Vasiliki Sakellari

**Affiliations:** 1Physiotherapy Department, School of Health and Care Sciences, University of West Attica, Agiou Spidonos 28, 2243 Athens, Greece; mpapad@uniwa.gr; 2Department of Physiotherapy, School of Health Rehabilitation Science, University of Patras, 26504 Rion, Greece; lampropoulou@upatras.gr; 3Laboratory for Research of the Musculoskeletal System (LRMS), School of Medicine, National and Kapodistrian University of Athens, University Campus, 15784 Athens, Greece; galanostat@yahoo.gr; 4Gait & Motion Analysis Center, ELEPAP, Kononos 16, 11634 Athens, Greece; gait_lab@elepap.gr (G.G.); mtziomaki@gmail.com (M.T.)

**Keywords:** functional, treadmill training, partial body weight support, children, ataxia

## Abstract

**Background/Objectives:** Ataxia is quite common in pediatric neuromotor disorders and has a highly heterogeneous etiology. Mobility difficulties and functional limitations reflect the lack of coordination in this population. The aim of this study is to assess the effectiveness of an intensive program of Functional Partial Body Weight Support Treadmill Training (FPBWSTT) on the mobility and functionality of children with ataxia. **Methods:** Through a stratified randomized control trial, a sample of 18 children with progressive and non-progressive ataxia and GMFCS II-IV (mean age: 14 years; standard deviation: 2.5) was assessed prior to the intervention, post-intervention, and 2 months after its end. Motor and functional skills were assessed with the Gross Motor Function Measure (GMFM, items D-E), the Pediatric Balance Scale (PBS), a 10 m walk test (10 MWT), a 6 min walk test (6 MWT), the Scale for Assessment and Rating Ataxia (SARA), the TimedUp and Go (TUG) test, spatiotemporal gait parameters, and kinetic and kinematic variables of the pelvis and lower limb. **Results:** Statistically significant interactions and changes in favor of the FPBWSTT were found in all functional assessments and spatiotemporal gait parameters (*p* < 0.05), the majority of which were maintained for two months. There was no statistical interaction or change in kinematic parameters (*p* > 0.05), while kinetic variables were insufficiently collected and were not statistically analyzed. **Conclusions:** The FPBWSTT is more effective on the mobility and functionality of children with ataxia who are 8–18 years old, compared to typical physiotherapy. Kinematic variables may not be sensitive indicators of change over a short period of time and/or in this population.

## 1. Introduction

Ataxia in childhood is a clinical manifestation of neuromotor disorders, the occurrence of which is estimated at about 26/100,000 children in Europe and probably reflects a minimum prevalence worldwide [1]. In many countries, including Greece, official data on the prevalence of ataxia in the pediatric population are lacking, while both assessment and intervention strategies seem to have been poorly studied worldwide [2].

The pathogenesis of childhood ataxia is characterized by great heterogeneity, with progressive and non-progressive cerebellar ataxia being its two main types [3]. The causes of this movement disorder can be acquired, such as cerebellar tumors; congenital, such as cerebral palsy; or genetic, such as Friedrich’s ataxia [3]. European data show that the highest incidence of ataxia in children is overall ataxia of genetic origin (14.61/100,000 births), while ataxia as a result of CP is the most studied in Europe and the most frequent among non-progressive disorders (10.65/100,000 births) [1]. In accordance with European data, a recent Greek cross-cultural adaptation [4] of the Scale for Assessment and Rating Ataxia (SARA) [5] found a rate of 54.4% for ataxic cerebral palsy and a rate of 62.5% for Friedrich ataxia in the non-progressive and progressive ataxia categories, respectively.

The symptoms of a taxi are ferto a set of difficulties that manifest in each type and etiology of ataxia, with varying degrees of severity and limitations in daily activities [6]. Lack of balance and coordination are the primary difficulties, evident in the quality and quantity of daily activities where children interact. Ataxic gait, dysmetria, dysdiadochokinesia, tremor, nystagmus, and dysarthria are some of the key clinical features underlying the symptomatology of cerebellar ataxia, regardless of its etiology [7,8,9]. According to Hartley et al. [2], investigating the assessment and treatment of children with various types of ataxia, it appeared that over forty different outcome measures were used in the twenty studies included in their systematic review. The majority of these studies focused on balance, gait, and gross motor function, identifying gait performance as a basic physiotherapy goal. The few data existing from children with ataxia report reduced cadence, step, and stride length; increased step width; variability in gait measures; and abnormalities in motor and kinematic characteristics [10,11,12,13]. These factors may serve as sensitive indicators of the progression of ataxia but also indicate the “instability” of ataxic gait pattern within the heterogeneous population investigated [10,11,12,13]. However, it seems that changes in gait performance and parameters in the pediatric population with ataxia after physiotherapy interventions have been poorly studied, mainly within short-term frames [2].

Treadmill interventions with and without partial body weight support are suggested as a means of faster gait acquisition, gross motor and gait parameters’ improvement in pediatric populations with various neuromotor disorders. However, the application of different protocols in different neuromotor populations, coupled with the small sample sizes, fail to demonstrate a clear and effective guideline [2,14,15,16,17]. An important factor of the therapeutic effectiveness is the dosage of training, which varies according to the intensity (e.g., treadmill speed, incline) or duration [17], prompting the need to standardize and test protocols appropriately designed for specific pediatric populations. The strategy of increasing speed during gait training has been well documented with positive results, but speed alternations, as suggested by Bjornson et al. [18], for children with CP, are so far a rather “challenging” approach, worthy of investigation, especially in ataxia. The cerebellum, the main dysfunctional structure resulting in ataxia, has a regulatory role in gait speed control, and its activation during such challenging gait training [19] may be a promising therapeutic intervention. Another suggested therapeutic strategy recorded in adults with neuromotor deficits [20,21] and pediatric populations [22,23] seems to be dual-task training, which defines a functional and demanding approach, especially when combined with gait, capable of promoting neuroplasticity [24].

Through this study, we aim to investigate the effects of integrating speed-alternation treadmill training with dual-task functional activities on improving gait performance and gross motor function in children with ataxia. We anticipate that this study will provide further clinical, practical, and research insights, as well as a better understanding for future therapeutic purposes.

## 2. Materials and Methods

A stratified, randomized, controlled trial of 4-week Functional Partial Body Weight Support Treadmill Training (FPBWSTT), with a 2-month follow-up period, was applied to children with ataxia (progressive and non-progressive), aged 8–18 years and Gross Motor Function Classification System (GMFCS) II-IV. The clinical trial was registered to ISRCTN (ISRCTN54463720), following ethical approval from the University of West Attica (study protocol: 14η/26 April 2021) and ‘ATTIKON’ General University Hospital of Athens (study protocol: Γ ΠAΙΔ, ΕΒΔ 149/20 March 2020). Participant recruitment started in June 2021 and was finally completed in the middle of December 2022, with the last data collected in the end of March 2023.

The protocol of this study with all methodological information was applied as was published in March 2022 [25]. The consort checklist is further provided as a Appendix A.

### 2.1. Group Allocation

Inclusion and exclusion criteria for participant recruitment was administered as originally planned in the study protocol [25]. Participants were randomly allocated to the intervention group (FPBWSTT group) or control group, using a stratification method, based on the type of ataxia disorder. Two strata were used to stratify the sample: progressive and non-progressive ataxia type [26]. An independent researcher from data collection was responsible for creating the allocation process, which was kept hidden from the rest of the research team. Children from both groups continued to receive their typical therapy (physiotherapy and any other therapies they received, such as occupational therapy, speech therapy, etc.). The FBPWSTT group additionally followed the 4-week intervention program (FPBWSTT).

### 2.2. Intervention and Settings

The FPBWSTT was designed based on the implementation of a treadmill gait protocol with speed change intervals [18] combined with dual-task functional activities, according to applications in different population groups [20,21,27,28,29]. The implementation of functional motor activities in children with neuromotor disorders, as well as their progressivity, has been shown to be a potentially effective therapeutic strategy [30,31]. These activities were implemented in a cyclical rotation, as described in detail in the study protocol [25]. This approach creates a satisfactory repetition of challenges, during gait, to enhance performance [32]. The 4-week physiotherapy intervention has been studied both in children with cerebral palsy [18] and in children with ataxia [12], and this period seems to be a duration with beneficial therapeutic effects. A partial body weight support of <40% of each participant was applied through the use of LiteGait. This limit of body weight support has been suggested by the literature in order to improve gait performance in different types of gait disorder populations [14,33,34,35].

Twenty (20) sessions over 4 weeks (5 days/week) constituted the intensive FPBWSTT program. The session was divided into two phases. Phase 1 consisted of 20 min gait training on the treadmill, with no handlebars, and speed intervals of 30 s between low and high speed. Both low and high speeds were individualized and modulated at 75–80% of each child’s respective ground speeds according to the 10 MWT. The high speed was progressively increased per session up to 5% of the defined maximum speed achieved from the previous day (Figure 1). Phase 2 consisted of 15 min functional dual-task training. In this phase, functional activities were applied during the individualized fixed low-pace treadmill gait. In each session, 3 of a total of 6 activities were applied in a cyclical rotation, with each one lasting up to 5 min (Figure 1). In every session, each child’s performance was recorded. The physiotherapist encouraged the gradual improvement in motor performance in each activity with verbal and visual guidance. Participants wore their shoes throughout the intensive program, along with any insoles they might have used. At the completion of the two phases, a total of 35 min of gait (20 min with speed intervals and 15 min with dual-task functional activities) was completed and displayed on the treadmill monitor. The total time from preparation to completion and release of the child from the LiteGait system ranged from 45 min to 1 h “with breaks”.

The FPBWSTT, as well as the assessment of all participants’ spatiotemporal gait characteristics, was performed using a LiteGait LG 200 P (Mobility Research, Tempe, AZ, USA) partial body weight support system, accompanied by a Gait Keeper (GaitKeeper S22, Mobility Research, Tempe, AZ, USA) electric treadmill system with a low-gait start speed (0.1 km/h). Two tablets were used for monitoring, recording, and collecting data from LiteGait.

For the 3D gait analysis, two force plates, specially designed platforms (AMTIs), placed in the middle section of a 10 m path were used to obtain the kinetic data. For the kinematic data, 16 self-adhesive reflective markers were appropriately placed at anatomical points of the pelvis and lower limbs. These data were acquired from 10 Vicon cameras (6MXT10 and 4 Verov2.2) at 100 Hz and 2 Basler digital video cameras (Pentaxlenses, Basler IND CCD Pentax TV Lens 8.5 mm 1:15, Tokyo, Japan) at 50 Hz.

### 2.3. Typical Therapy

Participants in both groups continued to receive their regular therapeutic program, consisting of physiotherapy and any other form of therapy the child needed, such as occupational therapy, speech therapy, etc. Typical physiotherapy (co-counting therapeutic swimming or equine therapy) did not exceed a total of 3 days per week, as otherwise it could be considered an intensive physiotherapy program [36,37]. The typical physiotherapy included 45 min sessions 1–2 times per week with functional exercises aimed at improving standing and gait skills. These included exercises on the mat to strengthen the trunk and lower limbs, therapeutic Pilates exercise, standing exercises, maintaining an upright position with balance boards, transitions to and from the standing position, walking on the ground and/or on a treadmill, and climbing up and down stairs.

### 2.4. Measurement Outcomes

Demographic and anthropometric data, including the functional level of each child through the use of the Gross Motor Function Classification System (GMFCS) and data obtained by the Child quality of Life Questionnaire for children report were collected once in the baseline assessment [38,39]. Primary and secondary outcomes were assessed three times over a period of approximately 3 months. A first baseline assessment was performed prior to the 4-week intervention period (baseline), a 4-week post-assessment was conducted by the end of the 4-week intervention period (4-week), and a final 2-month post-assessment was conducted 2 months by the end of the 4-week intervention period (2 months) (Figure 2).

Primary Outcomes:The Gross Motor Function Measure (GMFM-88) dimension D/E (GMFM-D /GMFM-E), were expressed as percentage scores (%) for the assessment of the motor performance and functional ability in standing, walking, running, and jumping [40].

Secondary Outcomes:The Pediatric Balance Scale (PBS), with scores in a range of 0–56 (the higher score indicates better balance), was used for the assessment of functional balance skills [41].The 10-meter walk test (10 MWT) for the assessment of self-selected slow-(10 MWT-SLOW) and high-speed (10 MWT-FAST) gaits over ground and determination of individualized training speeds on the treadmill were expressed as meters/second (m/s) [18,42,43].The Timed Up and Go (TUG) test for the assessment of dynamic balance control was expressed in seconds (s) [44].The 6-minute walk test (6 MWT) for the assessment of physical condition and endurance was expressed in meters (m) [45].The Scale for Assessment and Rating Ataxia (SARA), applied in the Greek version (SARAgr), was used for the assessment of ataxia features, with scores in a range of 0–40 (a higher score indicates more severe ataxia) [4].Three-dimensional kinematic and kinetic analysis of lower limbs through motion and gait analysis was applied, with the electronic recording of kinematics elements for the pelvis, hip, knee, and foot of both lower limbs in the three planes of motion (sagittal, frontal, and transverse). Since laterality severity does not occur in ataxia, each participant’s dominant ankle power absorption and generation, expressed as Watt/kg, was collected for the kinetic variable. Similarly, the dominant lower limb’s mean gait deviations index (MGDI) during 1. pelvis movement in all three planes of motion; 2. hip, knee, and ankle movement in the sagittal plane; and 3. ankle movement in the transverse plane (foot progression) were collected and expressed as normally disturbed data, categorized as Normal Standard Deviations (NSDs) through Gait Deviation Index (GDI) analysis [46]. Data recording and collection were obtained by the Gait and Motion Analysis Lab of ELEPAP in Athens.Analysis of spatiotemporal gait features was performed through 2 min recordings on the treadmill with partial body weight support at a personalized slow walking speed (75% of each participant’s self-selected walking speed based on the 10 MWT over ground). GaitSens software (version 2.0), incorporated in the LiteGait equipment, was used for recording and collecting the gait parameters. The dominant lower limb’s step length, stride length, and width length were expressed in terms of meters (cm), while step and stride time were expressed in seconds (s).

### 2.5. Assessment Procedure and Raters

Assessments were conducted according to the suggested guidelines of the outcome measure creators, following the same conditions, the same equipment across all participants, and across all time points. All assessments were conducted barefoot, except for the 10 MWT, 6 MWT, and 2 min gait recording on LiteGait, in which participants wore their shoes, as these were not possible or safe to be assessed barefoot. The procedure to assess the spatiotemporal gait parameters on the treadmill was performed with minimal partial body weight support (0–15%) at the individualized low walking speed (75% of the walking speed chosen by each participant based on 10 MWT on the ground). Equipment placement on LiteGait was the same as in the intervention procedure. All the assessments besides the 3D gait analysis took place in the same private pediatric faculty, from the same physiotherapy team, which was not blind to each child’s allocation group.

For the 3D gait analysis, a total of 16 self-adhesive reflective markers were initially placed on participants at key anatomical points, according to the standard Plug-in-Gait biomechanical analysis model (Vicon, Oxford, UK) for the pelvis and lower limbs in order to collect kinematic data. Participants were instructed to walk on the 10 m path, completing a total of five barefoot walking trials in sequence, at their self-selected speeds, independently or with the help of a parent or physiotherapist, according to each child’s GMFCS level. Upon completion of five left and five right gait cycles, valid motor data were determined from the force platform, on which each leg of each participant had to be placed and on which each leg “landed” wholly and individually on the force plate. The recording and collection of the above kinematic and kinetic data (Figure 3) were performed by the Athens ELEPAP Gait and Motion Analysis Laboratory by a blind physiotherapy team.

### 2.6. Statistical Analysis

The proposed sample size estimation could not be verified with the completion of 5 children per group, as was initially scheduled [25], due to the large heterogeneity among the participants at that time period. Despite the fact that an interim statistical analysis was not planned in the design of the protocol, in March 2023, with available data from a total of 18 children with ataxia, calculation of the primary end point of the GMFM-D was performed. This indicated statistically significant differences, which safely led to the adjustment of the sample size and therefore the termination of further sample search [47]. The variables were described using the mean and standard deviation or the median and interquartile range (in cases of violation of normality). The Kolmogorov–Smirnov test was used to check the normal distribution of the data. A comparison for homogeneity between the intervention groups in relation to the demographic and clinical indicators was performed using the independent samples *t*-test and the Chi-square test.

A two-way mixed ANOVA model was used, considering “intervention” (between groups) and “time” (within group) as factors for the analysis of the variables, applying the Bonferroni correction for all pairwise comparisons either between the intervention groups or between the time assessments.

Sensitivity analysis of the variables concerning the homogeneity of the groups at baseline (baseline–balance) was performed using two methods:The percentage change from baseline at all time points, where we compared the percentage changes between the intervention groups using the independent samples *t*-test or the Mann–Whitney test if the data did not follow a normal distribution.The absolute change from baseline at all time points, where we compared the absolute change in the variables (dependent variable) between the intervention groups (factor) and the baseline assessment (covariate) using the Analysis of Covariance (ANCOVA) model.

All statistical analyses were performed using the SPSS statistical package, version 21.00 (IBM Corporation, Somers, NY, USA). All tests were two-sided. A *p*-value < 0.05 was considered statistically significant.

## 3. Results

### 3.1. Participant Characteristics

Thirty-five (35) children with ataxia were assessed according to the eligibility criteria. A total of 21 children were randomized to either the intervention [Functional Partial Body Weight Support Treadmill Training (FPBWSTT)] or the control group, as shown in Figure 4. The data from 18 children were finally collected and statistically analyzed, from which only 9 children completed the 3D gait analysis in all three assessment periods, due to time-restricted periods from the 3D gait lab. Those data were processed for statistical analysis. The two groups were similar at baseline, in all clinical and demographic features, as well as in their perception of quality-of-life issues, as shown in Table 1.

### 3.2. Primary Outcomes: Domains D and E of the GMFM-88 (%)

The FPBWSTT group was statistically significant improved in GMFM domains D (standing) and E (gait) at 4 weeks by a mean of 6.45% (95% CI 3.34–9.57, *p* < 0.001) for standing and by a mean of 4.68% (95% CI 1.66–7.70, *p* = 0.005) for gait, compared to the control group. These improvements were also maintained at 2 months, by a mean of 5.56% (95% CI 1.46–9.67, *p* = 0.011) and by a mean of 8.51% (95% CI 2.91–14.11, *p* = 0.005), compared to the control group, respectively. Gross motor functionality of standing and gait changed significantly through time for the FPBWSTT group (*p* < 0.005), but no significant changes were found for the control group (*p* > 0.05). This is reinforced by the significant interaction of the groups and time for both variables (*p* = 0.002), as shown in Figure 5.

### 3.3. Secondary Outcomes

Analyses of the data of secondary outcome measures are presented in Table 2,showing that in most of them, the FPBWSTT group had statistically significant improvement (*p* < 0.05) at 4 weeks, compared to the control group. The intervention group maintained their improvements in the PBS, 6 MWT, and spatiotemporal gait parameters at 2 months, as shown in Table 2 and Figure 6. There were baseline differences between the groups, both in stride and step length, with greater values obtained in the control group (Table 3). Interestingly, the slow self-paced 10 MWT did not show statistically significant improvement at 4 weeks (mean: 0.07 m/s, 95% CI −0.04/0.18, *p* > 0.05) and at 2 months (mean:0.05 m/s, 95% CI −0.05/0.15, *p* = 0.307), for the FPBWSTT group. This was in stark contrast to the fast self-selected gait pace at 4 weeks, in which the intervention group showed statistically significant improvement, by a mean of 0.19 m/s (95% CI 0.00–0.38, *p* = 0.046).

The percentage change of primary and secondary outcomes after 4 weeks and 2 months of all participants is presented, respectively, in Figure 6 and Figure 7.

#### Secondary Variables from 3D Gait Analysis

Three-dimensional gait analysis was obtained only in a total of 9 children (FPBWSTT group *n* = 5; control group *n* = 4). Kinetic variables were inadequately collected, due to the insufficient placing of the entire foot or due to the coexistence of both feet on the platform during gait.

The values of the kinematic variable of the pelvis in the sagittal plane were highly heterogeneous with extreme values. Additionally, given the small sample size, the statistical analysis of this variable was excluded, as no valid result could be attributed [48]. The kinematic variables that were analyzed were the ones obtained from 1.pelvis movement in the frontal and transverse planes of motion; 2. hip, knee and ankle movement in the sagittal plane; and 3. ankle movement in the transverse plane (foot progression). Those findings indicated no statistically significant differences between the groups and no statistically significant changes for either of the groups, as shown in Table 4.

## 4. Discussion

The results of the clinical study, alongside compliance to the protocol and the absence of adverse effects, indicate that the FPBWSTT can be a highly effective intensive program for children8–18 years old with moderate functional severity of ataxia, more than typical physical therapy. The significant improvements in gross motor function in the areas of standing and walking were accompanied with similar changes in ataxia symptoms, functional balance skills, gait speed, dynamic balance control, physical condition and endurance, and spatiotemporal gait parameters, in favor of the FPBWSTT group. The majority of these were sustained two months after the end of the intensive program. The collection and statistical analysis of the kinematic variables of the 3D gait analysis did not reveal significant differences between groups, and neither significantly changed in any of the two groups between the different time measurements. The sample that completed the 3D gait analysis was too small (only nine children), which limits the statistical power for these specific measures and may affect the robustness of the conclusions drawn from these data. However, these findings may indicate that such measurements may not be sensitive enough for detecting functional changes in the specific population and/or at such a short time frame.

Children in both groups had similar clinical and demographic features. Confounding factors that could affect the outcome, as well as the interpretation of the results, were taken into account. This was obtained through the admission and exclusion criteria of the participants, as well as their recruitment, based on stratification of the type of disorder. Each group had one participant with progressive type of ataxia and GMFCS IV, while the remaining participants belonged to the non-progressive ataxia category, between GMFCS II-III. Children who had undergone surgical resection of the posterior fossa tumor (neuroblastoma), as well as the child with ataxia as a result of traumatic brain injury, all far exceeded the time frame for spontaneous recovery of balance ability, determined to be up to one year [49].

According to the results in gross motor function, particularly in standing and walking after the end of the of the present study, the children who completed the FPBWSTT significantly improved intensive program, which were maintained at follow-up. At the same time, statistically significant differences emerged between the two groups in the changes from the initial measurement, both after the end of the 4-week period and after the 2-month period. The domains of standing (domain D) and walking (domain E) in gross motor function, according to the GMFM, are the most representative domains of motor difficulties in children and youth with CP [50,51], as well as in children with ataxia [2,12].

The findings of the present study are even more encouraging than the reference range of 0.8–5.2% and 2.3–6.5% suggested by Storm et al. [52] as a Minimum Clinically Important Difference (MCID) for the improvement of GMFM-D in children with cerebral palsy and TBI, respectively, after one month of robotic gait training. Accordingly, the reference range of clinically significant difference for GMFM-E varies from 0.3 to 4.9% for children with CP and 2.8–6.5% for children with TBI. Specifically, in the study there was a total of 182 children, aged 4–18 years with movement disorders as a result of CP and TBI, who followed 20 sessions of 45 minutes of gait training and typical physical therapy each (total 90 min session) over a 4-week period [52].

In the present study, the change in gross motor function in standing position after the end of 4 weeks was 6.81% for FPBWSTT group and only 0.35% for the control group, while the change in gross motor function in gait was 5.30% for FPBWSTT and only 0.62% for the control group. Those findings are in accordance with the pilot study of Peri et al. [12], which investigated the effects of a 4-week gait training program in 11 children with ataxia due to TBI, additionally reinforcing the clinical significance of gross motor function change in standing and gait domains in such a short time frame. An interesting point found in the present study and confirmed through the research of Peri et al. [12] is that both programs focused on gait training produced greater changes in standing gross mobility than in gait. In other words, what is evident is that even if there are no exercises focused on static and dynamic standing skills, gait training contributes significantly to the enhancement of gross motor function in standing.

The exact same trend in change, similar to GMFM-D, was found for the PBS, with a clinically significant change, based on the proposed reference range after a 6-month period for children up to 6.5 years old with CP [53]. The same study showed a strong correlation between the total score of GMFM-66 and PBS. Based on our findings, GMFM-D and PBS share common characteristics with a trend change consistent with basic principles of neuroplasticity [33], as practicing a motor activity promotes a type of transfer to the acquisition and improvement of similar activities to the one being trained. It appears that gait training, as a functional activity, and the way in which speed alternations and dual-task functional activities were combined, were able to promote simultaneous or subsequent changes, through a network of neural circuits for other functional activities, in addition to those of gait. Motor training alone is capable of activating angiogenic mechanisms in the motor cortex and cerebellum and promoting the growth and survival of neurons in many brain regions [54,55]. The activity of gait involves activation of somatosensory and motor areas of the whole body, and the application of FPBWSTT through both speed and dual-task challenges appears to be sufficiently able to promote the appropriate synaptic connections and form a fertile environment to support structural brain changes related to standing functional activities.

A statistically and clinically significant difference was also found for gross motor function in gait (GMFM-E), with an interesting, continuous improvement observed 2 months after the end of the FPBWSTT. This finding could be due to a possible increase in the relevant daily activities of the children who completed the FPBWSTT as a “habit” formed as a result of the intensive gait program. As supported by theories and research on motor learning, neuroplasticity is an ongoing process, rather than a single event [54].The consolidation of a motor behavior involves time and repetition that often depend on the time period after the end of the motor training [54], which could explain the continuous increase in GMFM-E, 2 months after the end of the training. Unfortunately, this remains an assumption, based on theoretical knowledge that can not be directly correlated with other studies, which investigated short-term and post intervention results or the total GMFM score of all domains [2].

The change in ataxic symptoms was only significant just after the end of the 4-week program for the FPBWSTT group, without reaching the minimum clinically important difference reported in adult studies [56,57] and in the pediatric population [12,58]. It is possible that further clinical reduction in the SARA score can be achieved by a more targeted therapeutic approach on functional activities, related to the ataxic signs and increased time practice, as the cerebellum, the main responsible structure causing ataxia, requires sufficient temporal processing [55].

Improvements in dynamic balance control following various applications of therapeutic protocols have been reported in children with ataxia with reference to fall risk reduction and functional mobility skill improvement [2]. However, the heterogeneity of study protocols in the measurement of TUG and the need for chronological age correlation with a standard reference value [59] restrict the clinical interpretation of TUG’s change seen in the present study. Nevertheless, the improvement and significant change found for the FPBWSTT compared to the control group at 4 weeks indicates developing functional independence in children [60]. This is reinforced by the improvements in the self-selected pace and the significant change in10 MWT-fast, which further points out the ability of adjusting and regulating gait speed—an extremely difficult task for people with cerebellum deficits [19]. Besides the above, the increase in gait speed found for the FPBWSTT is even more apparent in the way that 6 MWT changed. Even though 6 MWT measures physical condition and endurance, it has been widely used informally for walking speed calculation in children with CP [61], while a strong correlation with 10 MWT has been supported in adults with neurological deficits [62]. Based on the increased 46, 25 m by the end of the FPBWSTT, a study similar in population, duration, and type of intervention [12] identified an improvement of 48 m, post 4-week gait training. These findings may verify an important reference range for the functional improvements of children with ataxia, after 4-week, 20-sessiongait training.

Children who underwent the FPBWSTT significantly reduced their step width at all time periods, showing statistically significant changes, compared to children who underwent a standard program. Despite the fact that the children of the FPBWSTT group had significantly shorter stride and stride lengths compared to the control group at baseline, there was a statistically significant increase in their stride and step length after 4 weeks, in contrast to children of the control group who even demonstrated a decline in those gait variables. However, from the absolute changes, only that of the step length at 4 weeks was statistically significantly improved. The increase in stride length continued further, even 2 months after the end of FBPWSTT, while step length showed a slight decrease. This may be due to an increased variability of steps with non-symmetrical bilateral steps at the given time point. In a systematic review by Buckley et al. [10], regarding gait characteristics in adults with ataxia, increased variability of stride length, but not of step length, was reported at low speeds compared to higher speeds. Given the above, this finding could be due to the difficulty of adapting gait to a walking speed lower than what has now been achieved, as the speed of evaluation of all spatiotemporal characteristics in all three time periods was the same (75–80% of low self-selected baseline ground speed).

The overall spatiotemporal gait changes are of particular importance, as they highlight an adapting ability and the possibility of adopting a new gait strategy within the same gait speed. Increased step and stride time characterize ataxic gait, compared to typical values of adults and children [10,12]. According to the present findings, what theoretically appears as worsening actually reveals adaptation of a gait pattern in order to maintain the same pace. It seems that temporal gait variables act as a compensatory index to maintain walking pace when spatial parameters (step and stride length) are increased as a result of the FPBWSTT program. This adaptation skill is a well-established indicator of improved motor functionality [63].

### 4.1. Kinematic Changes

According to current data, joint mobility in children with ataxia has been assessed in relation to the maximum trajectory range achieved during gait, expressed in degrees, compared to a healthy population [12,13] and immediately after gait training [12]. The assessment of the maximum range of motion at a joint and whether this is close to standard reference values is not necessarily an indication of improvement, as it is unknown whether this range of motion is utilized at the appropriate time during gait. The use of the MGDI seems to reflect the deviation between the pathological and typical gait, as it is calculated in such way as to estimate and compare the absolute value of each momentary deviation with the corresponding value of normal data [64].To our knowledge, the present study is the first in this population to assess changes in mean joint mobility deviations throughout the gait cycle, as derived from standard gait analysis values, relative to a control group and in the long term.

During 3D gait analysis of the nine children, there was an inability to collect kinetic variables due to incorrect and insufficient placement of the foot on the force-recording platform. Additionally, there was great heterogeneity with extreme values in the kinematic variables of the pelvis in the sagittal plane, an observation confirmed in adults with ataxia [65,66]. The instability of the trunk and pelvis in all directions of movement and especially in the sagittal plane, seems to be particularly pronounced in adults with ataxia, perhaps due to an effort to maintain an energy-efficient gait [65,66]. In a similar study by Peri et al. [12], changes in pelvic kinematics, measured by maximal range of motion after 4-week gait training showed greater values in deviation from the normal reference ones and increased standard deviation, following a different course of change, than the otherwise functional improvements of the participants. It seems that pelvis kinematics’ deterioration or even variability may be part of an effort for trunk stability in a newly adopted gait strategy.

All other kinematic variables did not show a specific change trend in any of the two groups, which differs from the findings reported by Peri et al. [12]. The latter argue that changes in knee and ankle mobility, close to standard values, may be indicators of the effectiveness of a therapeutic program. The pilot study of Peri et al. [12] applied 20 sessions of typical physiotherapy alongside with the 20 sessions of gait training, which aimed to improve balance skills and correct the movement pattern. On the other hand, in the present study, neither the typical physiotherapy nor FPBWSTT, specifically focused on “correction” or facilitation techniques, aimed to improve joint mobility. Changes in some kinematic variables could be expected over a longer time period, perhaps as an indication of a “consolidated” gait, without clarifying whether these changes will align with the reference standard values. What needs further clarification is whether these variables are associated with functional improvements in this population and whether physiotherapy should aim for “correction” of those.

### 4.2. Limitations

From the study design, a limitation is the lack of blind data collection by the assessors of the outcome measures, conducted at the private pediatric center, as they were also FPBWSTT therapists. The only blind assessors were the ones from the 3D gait analysis. This can be appropriately anticipated in future research, and the risk of bias arising can be therefore reduced.

The 2-month follow-up may not fully capture the long-term impact of the intervention, so future studies should involve a longer follow-up period to better evaluate the treatment effects over time.

## 5. Conclusions

In conclusion, our study provides robust evidence that FPBWSTT is a feasible and effective intervention for improving mobility and functionality in children with ataxia. Specifically, children in the FPBWSTT group exhibited a statistically significant improvement in gross motor function, with a 6.45% increase in standing performance (GMFM-D, *p* < 0.001) and a 4.68% increase in gait performance (GMFM-E, *p* = 0.005) after the 4-week intervention. These improvements were maintained at the 2-month follow-up, as demonstrated by significant differences between the intervention and control groups (*p* < 0.05). In addition, significant enhancements in dynamic balance and endurance, as measured by the Pediatric Balance Scale and the 6-Minute Walk Test, further support the efficacy of the FPBWSTT program and its maintained benefits over a period of 2 months. Interestingly, significant changes in some gait parameters indicate the adapting ability and the possibility of adopting a new gait strategy with the completion of the FPBWSTT. Although the kinematic variables did not show significant changes—likely due to a limited sample size—the observed functional improvements are both statistically and clinically meaningful. These results underscore the real-world applicability of FPBWSTT in pediatric rehabilitation settings, suggesting that this intervention can substantially enhance daily functional activities and overall quality of life for children with moderate ataxia.

## Figures and Tables

**Figure 1 jfmk-10-00123-f001:**
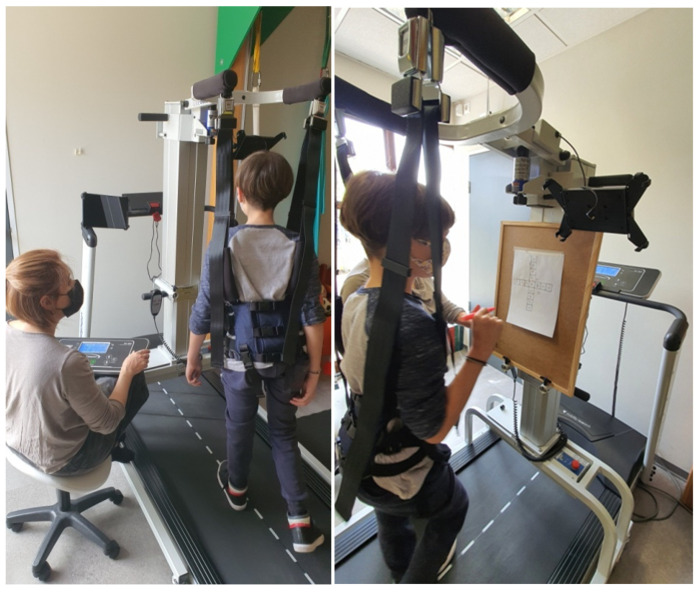
Application of the FPBWSTT. The left image shows gait training with speed intervals (phase 1), and the right image shows the application of the dual-activity of “target demonstration”(phase 2).

**Figure 2 jfmk-10-00123-f002:**
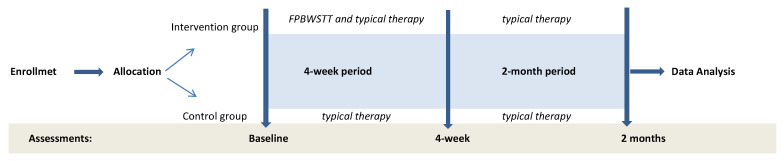
Flow of the events for the participants of this study.

**Figure 3 jfmk-10-00123-f003:**
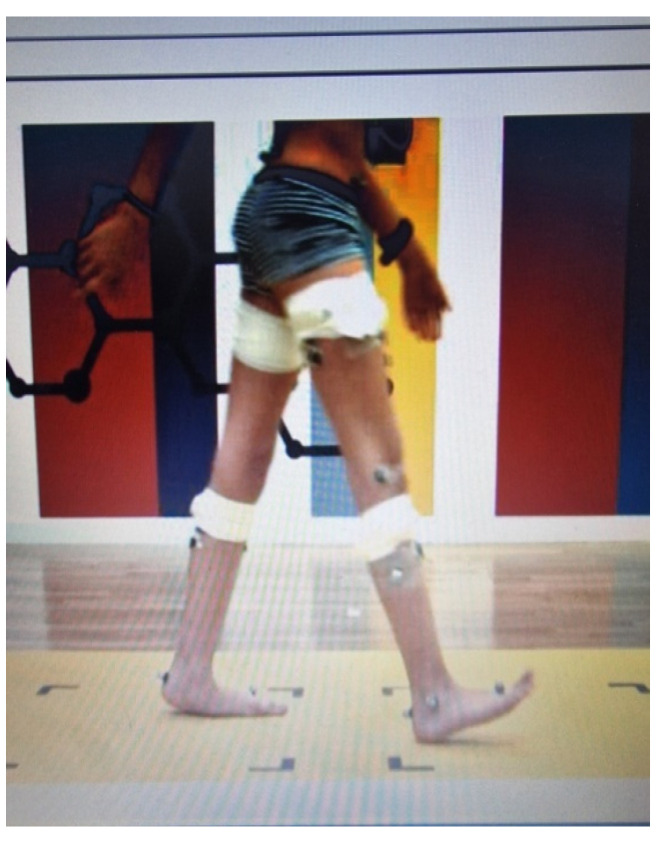
Gait procedure in a 10 m walking path during the 3D gait analysis in ELEPAP, Athens.

**Figure 4 jfmk-10-00123-f004:**
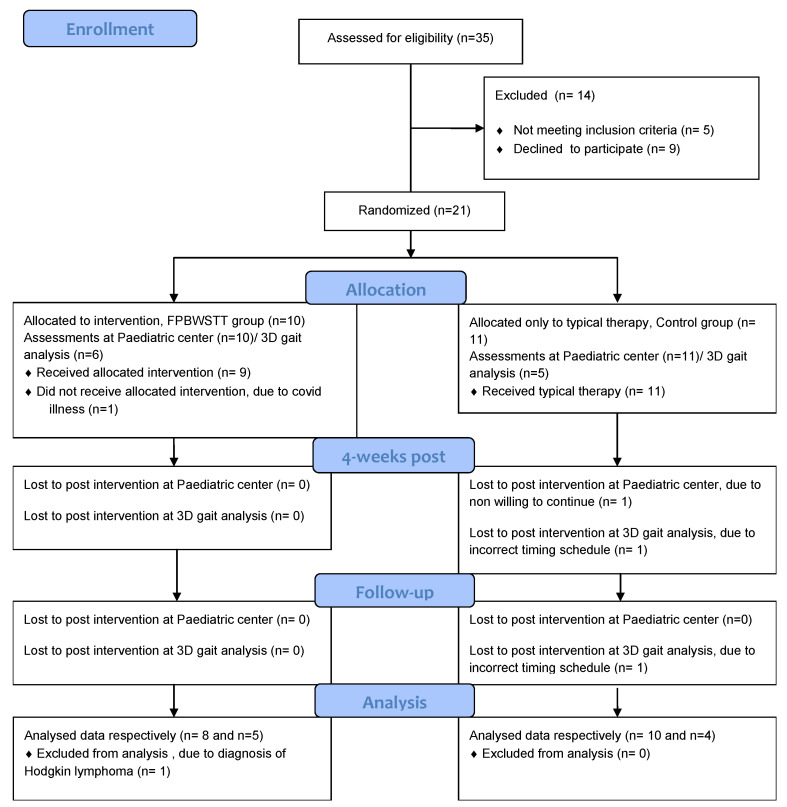
Flow chart of the study according to the Consolidated Standards of Reporting Trials “CONSORT” guidelines; pediatric center: private faculty for the evaluation of all the assessments; 3D: three-dimensional gait analysis at ELEPAP, Athens.

**Figure 5 jfmk-10-00123-f005:**
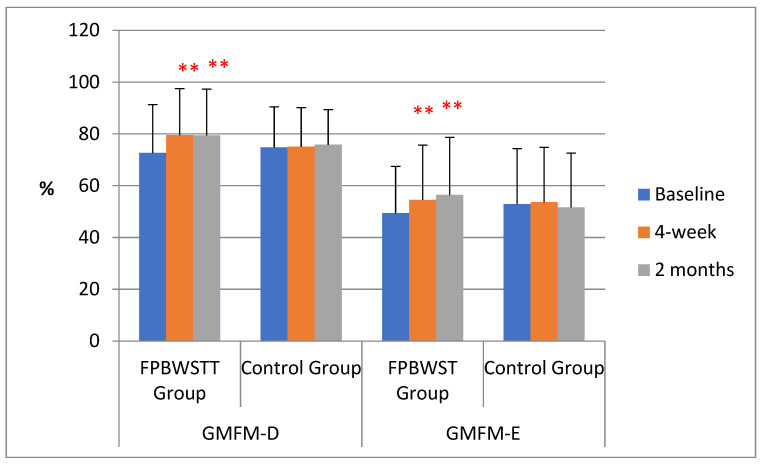
Mean values and standard deviation of gross motor function in standing (GMFM-D) and gait (GMFM-E) of the two groups at baseline, 4 weeks, and 2 months. Asterisks (**) represent statistically significant values, with *p* ≤ 0.005.

**Figure 6 jfmk-10-00123-f006:**
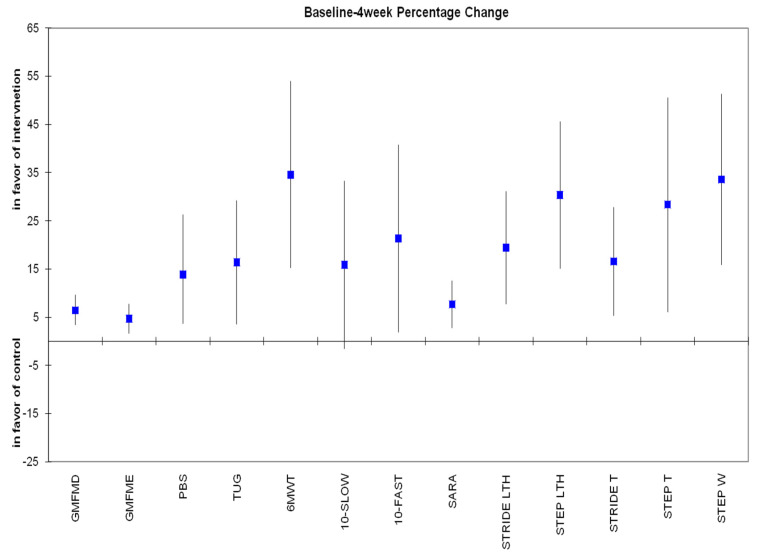
Percentage change from baseline to 4 weeks of all variables in which all 18 children were assessed. All changes are statistically significant (*p* < 0.05), in favor of the experimental group, except for 10 MWT-Slow, which is marginally statistically significant (*p* = 0.072).

**Figure 7 jfmk-10-00123-f007:**
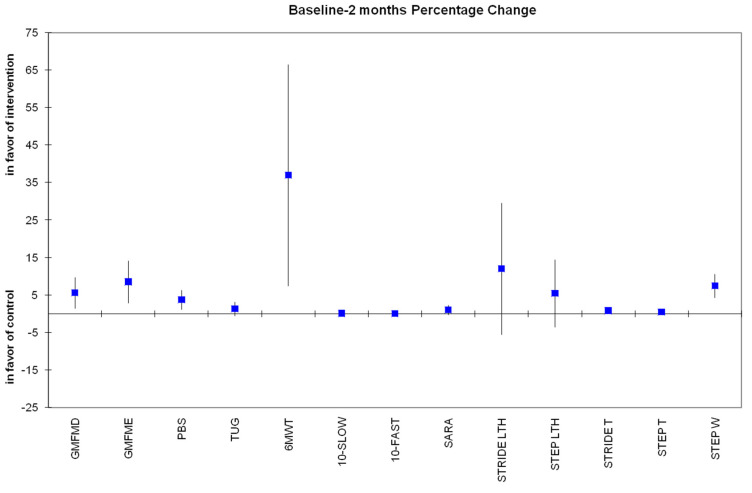
Percentage change from baseline to 2 months of all variables in which all 18 children were assessed. Most changes were still statistically significant (*p* < 0.05), in favor of the experimental group, except for TUG, 10 MWT-Slow, 10 MWT-Fast, and SARAgr.

**Table 1 jfmk-10-00123-t001:** Comparison of demographic, clinical, and quality-of-life characteristics between groups.

Characteristics	FPBWSTT Group	Control Group	*p*-Value
Age (years), mean ± SD	14.69 ± 2.05	13.45 ± 2.73	0.304
Sex, male/female	7 (87.5%)/1 (12.5%)	7 (70%)/3 (30%)	0.588
Weight (kg), mean ± SD	41.56 ± 8.79	46.55 ± 14.53	0.407
Height (cm), mean ± SD	157.38 ± 11.17	151.90 ± 14.67	0.397
BMI, mean ± SD	16.86 ± 2.20	19.72 ± 3.56	0.058
Type of ataxia, (non-P/P)	7 (87.5%)/1 (12.5%)	9 (90%)/1 (10%)	1.000
GMFCS, (II/III/IV)	6 (75%)/1 (12.5%)/1 (12.5%)	7 (70%)/2 (20%)/1 (10%)	0.909
Laterality, (right/left)	8 (100%)/0 (0%)	9 (90%)/1 (10%)	1.000
Quality of life Questionnaire			
Friends/Family, mean ± SD	7.18 ± 0.92	6.81 ± 0.82	0.375
Participation, mean ± SD	6.06 ± 2.31	5.65 ± 1.79	0.675
Communication, mean ± SD	6.58 ± 1.28	6.00 ± 1.50	0.395
Use of limbs, mean ± SD	5.99 ± 1.78	6.23 ± 1.62	0.683
Self-care, median ± IR	7.00 ± 0.0	7.00 ± 0.0	0.537
Equipment, mean ± SD	4.21 ± 1.38	4.21 ± 0.76	1.00
Pain-discomfort, mean ± SD	2.98 ± 0,77	2.98 ± 0.93	0.991
Sense of happiness, median ± IR	7.00 ± 1.0	7.00 ± 0.5	0.573
Assistance with questionnaire completion, mean ± SD	2.38 ± 1.06	2.40 ± 1.07	0.961

Abbreviations: SD, standard deviation; IR, interquartile range; BMI, Body Mass Index; non-P, non-progressive; P, progressive; GMFCS: Gross Motor Function Classification System. Note: Values are presented numerically, unless otherwise stated. There are no statistically significant differences between the groups at baseline (*p* > 0.05).Quality of Life Questionnaire: Scores are based on a Likert scale, from 1 (bad) to 9 (very good), except for the last question where the options range from 1 (no) to 4 (yes, very). Group Composition: **FPBWSTT Group** includes children with cerebral palsy ataxia (*n* = 5) [subsequent genetic testing revealed a variant of CACNA1A in 1 case], neuroblastoma (*n* = 1; 2 years post-surgical resection], CACNA1A-related ataxia (*n* = 1; previous CP diagnosis) and Friedreich Ataxia (*n* = 1). **Control Group** comprises children with cerebral palsy ataxia (*n* = 4), neuroblastoma (*n* = 1; 3.5 years post-surgical resection), CACNA1A-related ataxia (*n* = 1), ataxia of unknown etiology (*n* = 1), Gillespie syndrome (*n* = 1), TBI-related ataxia (*n* = 1; 6 years post-injury), and Menkes-like ataxia (*n* = 1).

**Table 2 jfmk-10-00123-t002:** Sensitivity analysis using ANCOVA model (baseline vs. 4-week and 2-month outcomes).

Secondary Outcome	4-Week Mean Difference (95% CI)	*p*-Value	2-Month Mean Difference (95% CI)	*p*-Value
PBS	3.73 (1.13–6.33)	0.008	3.85 (−0.14 to 7.71)	0.05
TUG (s)	2.19 (0.7–3.67)	0.007	1.29 (−0.53 to 3.10)	0.151
10 MWT-SLOW (m/s)	0.07 (0.04–0.18)	0.193	0.05 (0.05–0.15)	0.307
10 MWT-FAST (m/s)	0.19 (0–0.38)	0.046	0.04 (−0.12 to 0.21)	0.620
6 MWT (m)	56.09 (29.22–82.96)	<0.005	36.92 (7.34–66.50)	0.018
SARAgr	1.22 (0.44–2.01)	0.005	1.00 (−0.31 to 2.34)	0.125
Step Length (cm)	8.84 (1.67–16.01)	0.019	5.36 (−3.64 to 14.35)	0.224
Stride Length (cm)	8.29 (−3.98 to 20.57)	0.170	12 (−5.52 to 29.54)	0.165
Step Time (s)	0.38 (0.01–0.74)	0.043	0.45 (0.01–0.90)	0.048
Stride Time (s)	0.45 (0.16–0.75)	0.005	0.77 (0.05–1.49)	0.038
Step Width (cm)	6.74 (4.20–9.29)	<0.005	7.41 (4.18–10.64)	<0.005

Abbreviations: CI, confidence interval; PBS, Pediatric Balance Scale; TUG, Timed Up and Go; 10 MWT, 10 meter walk test; 6 MWT, 6 minute walk test; SARAgr, Scale for Assessing and Rating Ataxia (Greek version). Red highlights the statistically significant *p*-values.

**Table 3 jfmk-10-00123-t003:** Mixed two-way ANOVA for the step length and stride length variables.

Groups	Step Length (cm)			Stride Length (cm)		
	Baseline	4-Week	2 Months	Baseline	4-Week	2 Months
	Mean ± SD	Mean ± SD	Mean ± SD	Mean ± SD	Mean ± SD	Mean ± SD
FPBWSTT	22.85 ± 7.56	28.19 ± 8.69 *	27.49 ± 8.65	47.72 ± 12.74	54.79 ± 14.69	58.16 ± 18.66
Control	38.01 ± 7.04	35.99 ± 9.63	32.88 ± 7.00	72.29 ± 17.77	68.51 ± 18.27	64.57 ± 16.63
Comparison between groups by time	* p * < 0.005	*p* = 0.094	*p* = 0.163	* p * = 0.005	*p* = 0.104	*p* = 0.453
Interaction between group and time	F(2.32) = 6.94, *p *< 0.005			F(2.32) = 6.34, *p* = 0.005		

Abbreviations: SD, standard deviation. Asterisk (*****) indicates *p* < 0.05 vs. baseline. Red highlights the statistically significant *p*-values.

**Table 4 jfmk-10-00123-t004:** Sensitivity analysis of kinematic variables, using ANCOVA model (baseline vs. 4-week and 2-month outcomes).

Kinematic Variables	4-Week Mean Difference (95% CI)	*p*-Value	2-Month Mean Difference (95% CI)	*p*-Value
Pelvis Frontal Plane (NSDs)	−0.11 (−0.76 to 0.74)	0.972	0.57 (−0.42 to 1.57)	0.210
Pelvis Transverse Plane (NSDs)	−0.08 (−1.45 to 1.29)	0.888	−0.22 (−1.41 to 0.97)	0.671
Hip Sagittal Plane (NSDs)	−0.21 (−0.98 to 0.54)	0.512	−0.54 (−0.45 to 1.53)	0.232
Knee Sagittal Plane (NSDs)	−0.42 (−1.68 to 0.85)	0.450	−0.06 (−1.41 to 1.29)	0.919
Ankle Sagittal Plane (NSDs)	−0.21 (−0.70 to 0.28)	0.337	0.18 (−0.42 to 0.78)	0.482
Ankle Transverse Plane (NSDs)	−0.09 (−01.05 to 0.88)	0.829	−0.42 (−1.69 to 0.86)	0.452

Abbreviations: CI, confidence interval; NSDs, normal standard deviations.

## Data Availability

The data presented in this study are available on request from the corresponding authors.

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
