# Peer review of "Effects of Functional Partial Body Weight Support Treadmill Training on Mobility in Children with Ataxia: A Randomized Controlled Trial"

_jfmk, 2025, doi:10.3390/jfmk10020123_

Round 1

Reviewer 1 Report

Comments and Suggestions for Authors

This study evaluates the effects of functional partial body weight support treadmill training (FPBWSTT) on mobility and function in children with ataxia in a stratified randomized controlled trial. By integrating multiple functional assessments, including motor function, balance, and spatiotemporal gait parameters, the study provides a comprehensive understanding of motor improvements. Results show statistically significant improvements in motor function, balance, and gait parameters, some of which persist for two months. The study provides valuable insights into noninvasive, treadmill-based rehabilitation strategies for children with neuromotor disorders. However, I have several suggestions, comments, and questions that I'll include below.

  1. The title perfectly reflects the aim of the study, but it could be shorter and more attractive to the readership.
  2. The “Materials and Methods” subsection is uncharacteristically short. The methodology currently lacks sufficient detail on participant recruitment, intervention protocol, tools and equipment and statistical analysis. Everything is presented in a single paragraph, and this section could certainly be expanded to improve clarity and reproducibility. Meanwhile, the results section provides too many methodological explanations that would be better suited to the methodology section. Reorganizing these sections would make them easier to read and more consistent.
  3. The heterogeneity of participants is mentioned in the methodology section. Including both progressive and non-progressive ataxia cases introduces variability, making it more difficult to distinguish the effects of the intervention. It might be beneficial for the authors to acknowledge and analyze how participant heterogeneity may have influenced the results.
  4. The methodology should describe how kinematic data were collected (mentioned in the Results section, line 117 “9 children completed the 3d gait analysis in all three assessment periods“). The authors should explain the methodology of the kinematic measurements, including equipment, procedures, and analysis methods.
  5. The choice of statistical data evaluation methods is completely unclear. There are different types of data and they are all evaluated with two-way mixed ANOVA!? Justification and arguments for the choice need to be provided.
  6. Is a two-month follow-up sufficient to assess the long-term impact of an intervention? The study does not discuss whether the observed improvements are maintained beyond this period, which is essential for understanding the lasting effects of the treatment.
  7. Figures 2 to 5 need some kind of unified presentation. I recommend first adding a grid to all figures. Figures 2-3 do not have axis labels on the graphs themselves, this can only be judged from the title. Perhaps graphs 2 and 3 would look better if they were presented as bar graphs with value distribution intervals. The quality of graphs 4 and 5 is poor, difficult to follow and they differ greatly from other graphical presentations.
  8. The results are difficult to analyze due to the lack of a clear methodology presented beforehand. Methodological elements are mixed up with the results, and the presentation of graphs and tables is chaotic due to different representations, and font differences in both size and style.
  9. I advise you to avoid such a broad discussion because it gives the impression that the results obtained by others are analyzed in more detail than the authors' achievements presented in this article.
  10. In the conclusions, it would be good to emphasize the importance of the results for real-world application.

Although the study is valuable, there are still some shortcomings in this presentation, including the small sample size, heterogeneous participants, short follow-up period, and incomplete kinematic data. Further improvements to the manuscript should aim for greater methodological rigour to strengthen the conclusions.

Reviewer 2 Report

Comments and Suggestions for Authors

Dear authors,

Many thanks for the interesting manuscript. The authors examined the effect of a 4-weeks FPBWSTT intervention on the standing and gait gross motor functions (Domains D and E of the GMFM-88(%)) in children with ataxia. Overall, this is a very promising intervention idea. The results are also well discussed along the literature at the end. However, the necessary methodology is almost not mentioned at all and much is not introduced because reference is made to the existing paper with the same methodology. Without the first publication it is very difficult to understand this paper. The authors should introduce a minimum of the study design and procedure in this paper and explain important terms. A lot of abbreviations are used without having introduced them, for example.

Furthermore, a native speaker should definitely look over the document again. There are many sentence structures that are difficult to understand, punctuation errors and sometimes words are simply missing. You should also pay careful attention to spaces, e.g. between numbers and units or between text and citations. It is very inconsistent throughout the document. Also, some terms are spelled very differently in the manuscript (e.g. 4week, 4-week or 4 week). Please read the manuscript again carefully. Last but not least, pay attention to the use of hyphen, en-dash and em-dash. You mix them up.

I have corrected the pdf using the comment function, but the most important points are also listed below.

Introduction:

  • It is not clear from the background why a 4-week intervention period was chosen. The duration of the intervention can certainly make a difference. Could you please elaborate on this?
  • In the introduction, you state that you aim to investigate a novel functional intervention through the integration of different therapeutic approaches. However, you do not derive these new approaches in the introduction or clearly relate the cited approaches to your research question. The research question and hypotheses are too vague, making it unclear to the reader what exactly is being done. Please formulate this more clearly.

Methods:

  • Did both groups receive their usual therapy, with one group additionally undergoing FPBWSTT? Or was it an either-or approach? What do you mean by "usual therapy"? For comparing both groups, it is quite crucial to define it.
  • Even though there is already a study detailing the protocol, a brief summary of the tests should still be included here to ensure that this manuscript is understandable. The same applies to abbreviations. There are several tests and terms that are used in abbreviated form but are not introduced anywhere in this manuscript (e.g., 6MWT).
  • How did you check for normality and heterogeneity of variance? Please add the tests applied. If these criteria for an ANOVA were violated, how did you correct for it? Or why did you use an ANOVA despite the violation of these criteria?

Results:

  • Figure 2: It would be better if the significances were displayed in the image. Moreover, the ticks on the x-axis should align with the labels and the data points in the image. Furthermore, the unit of the y-axis is missing. The quality of the graphic is not high. This also applies to the following figures. Please improve the figures.
  • Tables:
    • A table must always be self-explanatory. This means that even if abbreviations have been introduced—which has not been done in this manuscript so far—they should still be spelled out in a footnote.
    • The footnote should also clarify the meaning of the red highlighting. Ideally, a distinction should be made between significant (p < 0.05) and highly significant (p < 0.001) in the marking, possibly using bold or *#.
    • The decimal places should always be consistent.
  • 3.1.: If the kinematic data could not be analyzed, why/how was a statistical analysis conducted? How is it possible that non-significances are reported here? And why did you not report the descriptive results in the text?

Discussion

  • Line 210-215: Without showing the data on which this statistical analysis is based or clarifying how many data points were actually included, it is difficult to make such statements. The sample size for this data analysis is likely far too small.
  • Please divide long sentences into several and check the grammar.

Many thanks and best regards

Comments on the Quality of English Language

A native speaker should definitely look over the document again. There are some sentences that are very long and complex. Please try to simplify and divide them up. Moreover, there are punctuation errors and sometimes words are simply missing. You should also pay careful attention to spaces, e.g. between numbers and units or between text and citations. It is very inconsistent throughout the document. Also, some terms are spelled differently in the manuscript (e.g. 4week, 4-week or 4 week). Please read the manuscript again carefully. Last but not least, pay attention to the use of hyphen, en-dash and em-dash. You mix them up.

Round 2

Reviewer 1 Report

Comments and Suggestions for Authors

The authors took into account all the comments submitted by the authors and significantly supplemented the publication. There is one more small nuance that would further contribute to the clarity of the publication - it is to support conclusions with significant results. Now the conclusions are just an opinion without confirmation.
I wish the authors success in their further work.
